# Integrin α5β1 nano-presentation regulates collective keratinocyte migration independent of substrate rigidity

**Jacopo Di Russo[1,2,3,4]\*, Jennifer L Young[1,5,6], Julian WR Wegner[1], Timmy Steins[2,4], Horst Kessler[7], Joachim P Spatz[1,8,9]\***

[1]Max Planck Institute for Medical Research, Heidelberg, Germany; [2]Interdisciplinary Centre for Clinical Research, Aachen, Germany; [3]DWI – Leibniz-Institute for Interactive Materials, Forckenbeckstrasse, Aachen, Germany; [4]Institute of Molecular and Cellular Anatomy, RWTH Aachen University, Aachen, Germany; [5]Mechanobiology Institute, National University of Singapore, Singapore, Singapore; [6]Department of Biomedical Engineering, National University of Singapore, Singapore, Singapore; [7]Institute for Advance Study, Department of Chemistry, Technical University of Munich, Garching, Germany; [8]Institute for Molecular System Engineering – IMSE - Heidelberg University, Heidelberg, Germany; [9]Max Planck School Matter to Life, Heidelberg, Germany

**Abstract** Nanometer-scale properties of the extracellular matrix influence many biological processes, including cell motility. While much information is available for single-cell migration, to date, no knowledge exists on how the nanoscale presentation of extracellular matrix receptors influences collective cell migration. In wound healing, basal keratinocytes collectively migrate on a fibronectin-rich provisional basement membrane to re-epithelialize the injured skin. Among other receptors, the fibronectin receptor integrin α5β1 plays a pivotal role in this process. Using a highly specific integrin α5β1 peptidomimetic combined with nanopatterned hydrogels, we show that keratinocyte sheets regulate their migration ability at an optimal integrin α5β1 nanospacing. This efficiency relies on the effective propagation of stresses within the cell monolayer independent of substrate stiffness. For the first time, this work highlights the importance of extracellular matrix receptor nanoscale organization required for efficient tissue regeneration.

\*For correspondence:
jdirusso@ukaachen.de (JDR);
spatz@mr.mpg.de (JPS)

**Competing interest:** The authors declare that no competing interests exist.

## Introduction

Collective cell migration is a fundamental biological process characterized by the coordinated movement of interconnected cells to achieve specific functions (*Ladoux and Mège, 2017*; *Rørth, 2012*; *Trepat and Sahai, 2018*). Basal keratinocytes collectively migrate to re-epithelialize a wound in the skin as the first step to re-establish tissue integrity (*Bornes et al., 2021*; *Safferling et al., 2013*). At the onset of migration, keratinocytes deposit a provisional basement membrane that provides the necessary support for adhesion required for cell locomotion (*Koivisto et al., 2011*). Among other components of this provisional matrix, fibronectin plays a pivotal role in supporting keratinocyte migration via the adhesion of α5β1 and αvβ1 integrins that are upregulated in re-epithelialization (*Cavani et al., 1993*; *Ffrench-Constant et al., 1989*; *Grose et al., 2002*). In contrast to αvβ1, which plays only a minor role in migration, α5β1 is required for efficient re-epithelialization, partially through the regulation of cellular traction forces (*Schiller et al., 2013*; *Zhang et al., 1993*). In epithelial collective migration, each cell needs to coordinate cell-matrix traction forces with cell-cell stresses in order to propagate directed migratory cues to surrounding neighbours (*Das et al., 2015*; *Ladoux and*

*Mège, 2017*; *Tambe et al., 2011*; *Vedula et al., 2013*). Previous work from our group and others have elucidated the many aspects of intercellular stress coordination governing collective cell migration (*Bazellières et al., 2015*; *Das et al., 2015*; *Sunyer et al., 2016*; *Tambe et al., 2011*; *Vishwakarma et al., 2018*), including the overall stress heterogeneity present in the migratory cell sheets, as well as the single molecular players involved in the mechanotransduction process (*Bazellières et al., 2015*; *Das et al., 2015*). Nevertheless, the role of extracellular matrix (ECM) adhesive sites in regulating such migratory coordination remains largely unknown.

Over the past two decades, it has become clear that nanometer-scale properties of the ECM strongly influence cell motility, in particular at the single-cell level (*Cavalcanti-Adam et al., 2006*; *Oria et al., 2017*). Previous work from our lab has highlighted the critical role of integrin lateral nanospacing in controlling single-cell spreading, migration speed, and persistence (*Cavalcanti-Adam et al., 2008*; *Cavalcanti-Adam et al., 2007*). Additionally, lateral adhesion spacing has been shown to regulate intracellular force generation in a manner consistent with the molecular clutch model (*Oria et al., 2017*). In order to understand the role of ECM nanoscale organization in collective cell migration, we took a bottom-up approach by synthesizing nanopatterned hydrogels of discrete spacings, such that adhesive site organization could be precisely controlled. In order to investigate integrin α5β1 regulation of keratinocyte re-epithelialization, we utilized an engineered α5β1 integrin-specific peptidomimetic as the adhesive ligand (*Figure 1—figure supplement 1*). This specific peptide has a high binding affinity for α5β1 integrin ($IC_{50}$ = 1.5 nM), but orders of magnitude lower affinity for the other RGD-recognizing integrins, including αv-containing isoforms (*Kapp et al., 2017*). The advantage of using synthetic hydrogels lies in their ability to better mimic the physical properties of the ECM in the wound, as well as in their protein-repellent nature. With this approach, we could systematically address the role of cell-ECM interactions in cell monolayers by controlling both ECM stiffness and ligand density with nanometer precision. We were able to show that keratinocytes require an optimal integrin α5β1 spacing in order to efficiently coordinate their collective movement independent from ECM stiffness.

## Results

### Integrin α5β1 surface density controls keratinocyte migration efficiency

To investigate the role of integrin α5β1 in regulating keratinocyte collective migration, we fabricated gold nanopatterns on glass surfaces via block-copolymer micelle nanolithography (BCMN). BCMN allows for the precise modulation of ligand density at the nanometer level. The nanopatterns were covalently transferred onto polyacrylamide (PAA) hydrogels of ~23 kPa stiffness (approximately the stiffness of freshly wounded skin) (*Arnold et al., 2004*; *Goffin et al., 2006*). The use of PAA hydrogels as culture substrate allows us to provide relevant substrate rigidity to cells and systematically control surface adhesion biochemistry due to the protein-repellent nature of PAA. Therefore, cells will only interact at nanoparticle sites, where highly specific integrin α5β1 peptidomimetics are linked, thereby allowing cell-surface adhesion via integrin α5β1 in a precisely controlled manner (*Mas-Moruno et al., 2016*; *Rechenmacher et al., 2013*; *Figure 1A*). In continuity with previous studies indicating cell-ligand length scale relevance for focal adhesion formation and cell motility, we modulated ligand density using three discrete inter-ligand distances: 35, 50, and 70 nm (*Cavalcanti-Adam et al., 2008*; *Cavalcanti-Adam et al., 2007*).

Confluent monolayers of human immortalized keratinocytes (HaCaT) were cultured within a lateral confinement using a polydimethylsiloxane (PDMS) stencil placed onto the nanopatterned PAA hydrogels (*Figure 1B*). Cell monolayer density and their proliferation rate were comparable among all three inter-ligand distances (*Figure 1—figure supplement 2*). After PDMS stencil removal, cell sheet migration was triggered onto the empty regions of the gel, mimicking the collective migration process of basal keratinocytes post-skin wounding (*Bornes et al., 2021*; *Safferling et al., 2013*). Quantifying cell sheet migration speed over culture time revealed a differential migration efficiency on the three conditions, with the highest speed observed at 50 nm (~20 μm/hr) vs. slower speeds at 35 and 70 nm (~10 μm/hr) (*Figure 1C and E*, *Video 1*). To exclude any specific behaviour connected to the HaCaT keratinocyte line, the same experiments and analyses were performed with human epidermis-derived keratinocytes (hKC) (*Sawant et al., 2018*) and with primary human epidermal keratinocytes (nHEK) (*Figure 1—figure supplement 3*). The speed quantification confirmed the higher migration

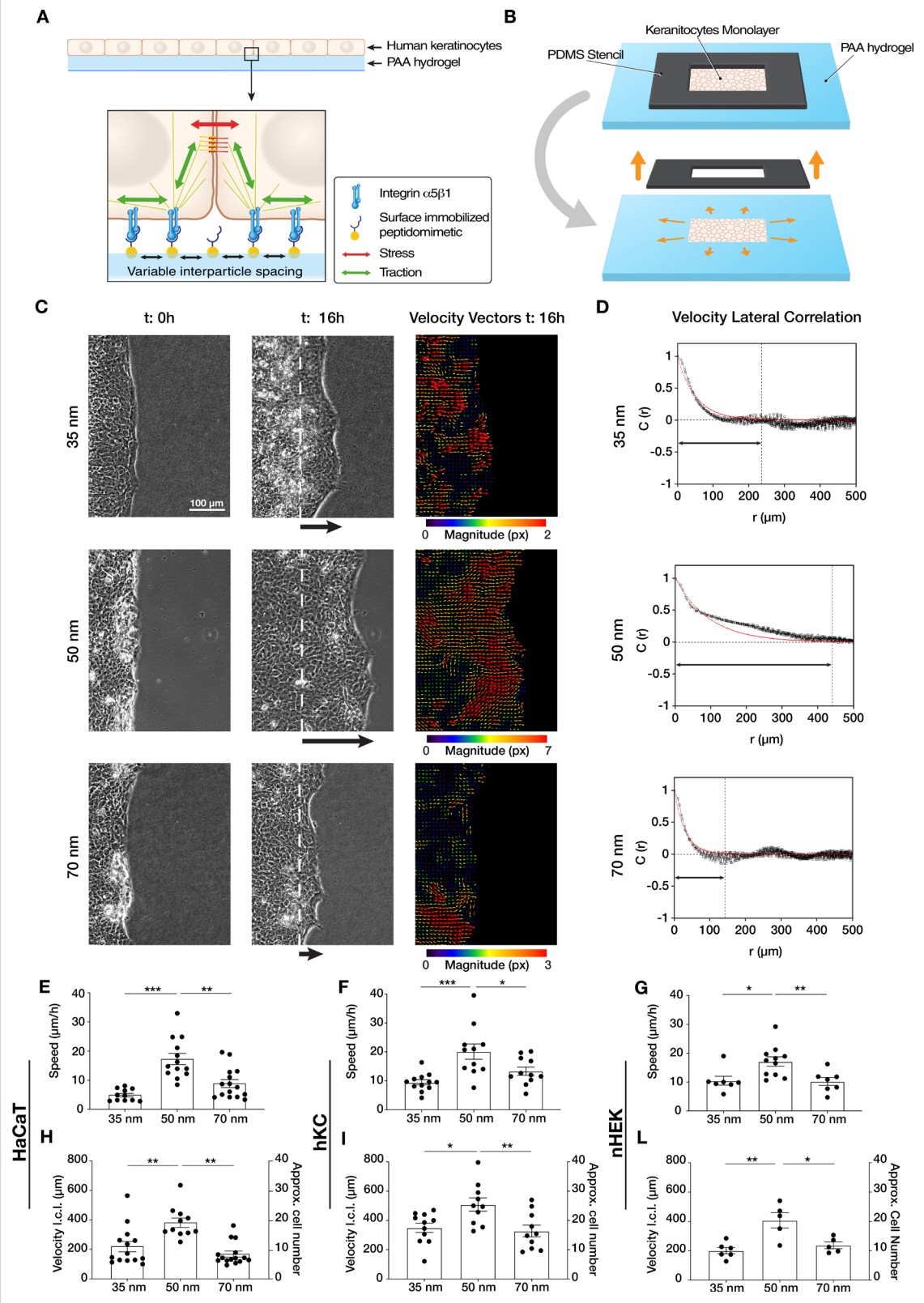

**Figure 1.** Efficient collective cell migration depends on integrin α5β1 ligand nanospacing. (**A**) Keratinocyte monolayers coordinate their intra/intercellular tractions/stresses on polyacrylamide (PAA) hydrogels nanopatterned with integrin α5β1 peptidomimetic. (**B**) Schematic representation of the migration experiment setup. (**C**) Representative images of time-lapse experiments at t = 0 hr (start) and t = 16 hr (end), with corresponding velocity vectors plots in pixel (px) of keratinocyte (HaCaT) sheet migration on hydrogels with 35, 50, and 70 nm integrin α5β1 ligand lateral spacing. The

*Figure 1 continued on next page*

*Figure 1 continued*

dotted white lines illustrate the initial starting point of the monolayers immediately following stencil removal, and the arrows indicate the direction of movement. (**D**) Representative velocity lateral correlation length, $C(r)$ where $r$ = distance, curves of keratinocyte (HaCaT) velocity vectors at $t$ = 16 hr. The quantification of migration speed and the lateral correlation length in HaCaT (**E, H**), human epidermis-derived keratinocytes (hKC) (**F, I**), and primary human epidermal keratinocytes (nHEK) (**G, L**) show an optimum at 50 nm integrin α5β1 ligand lateral spacing. Scatter plots show values with mean ± s.e.m. from at least three independent experiments. *p < 0.05; **p < 0.01; ***p < 0.001 using a Mann-Whitney test.

The online version of this article includes the following figure supplement(s) for figure 1:

**Source data 1.** Data points for graphs in *Figure 1* and its supplements.

**Figure supplement 1.** Chemical structures of integrin α5β1 selective peptidomimetic (**A**) and c(RGDfK) (**B**).

**Figure supplement 2.** Keratinocyte monolayer density and proliferation are not affected by integrin α5β1 nanopatterning.

**Figure supplement 3.** Human epidermis-derived keratinocyte (hKC) and primary human epidermal keratinocytes (nHEK) migratory behaviour is comparable with HaCaT.

**Figure supplement 4.** HaCaT keratinocyte collective behaviour on nanopatterned c(RGDfK) surfaces.

efficiency at 50 nm inter-ligand distance (*Figure 1F and G*), suggesting this is a universal characteristic of keratinocytes.

Next, we examined the extent to which ECM nano-presentation affects the directional migration of individual cells within the monolayer. Each cell in the monolayer coordinates its movement with its neighbours, and the length of such coordination can be quantified by comparing the lateral components (orthogonal to the direction of sheet migration) of the individual cell velocity vectors in the monolayer (*Das et al., 2015*). We thus used particle image velocimetry (PIV) to measure the velocity fields of migrating keratinocytes after 16 hr and calculated the lateral correlation length, $C(r)$ (*Figure 1C*). This revealed that keratinocytes migrating on 50 nm inter-ligand spacing most efficiently coordinate their movements, with an average correlation which extended over ~20 cells (~400 μm) (*Figure 1D and F*). In contrast, keratinocytes on both 35 and 70 nm inter-ligand spacing were less able to coordinate their movements, only propagating their movements across a few cells (~200 μm) (*Figure 1D and F*). Similar trends were also observed during hKC and nHEK migration (*Figure 1I, L*).

The density of integrin-ligand interactions alters cell spreading and migratory persistence through the regulation of focal adhesion maturation and dynamics in fibroblasts (*Cavalcanti-Adam et al., 2007*; *Cavalcanti-Adam et al., 2006*). Since the correlation length of epithelial cell migration can be regarded as single-cell migratory persistence, we analysed how integrin α5β1 lateral spacing influences focal adhesion structure and dynamics, as well as surface contact area of the cells (*Figure 2—figure supplement 1*). After transient transfection of HaCaT cells with an mCherry-α-paxillin construct, we tracked focal adhesions and quantified their lifetime and size (*Videos 2–4*). Cells exhibited the highest ability to remodel their focal adhesions when migrating on 50 nm-spaced α5β1 ligands, resulting in the fastest turnover, or lowest lifetime. In contrast, on 35 and 70 nm inter-ligand spacing,

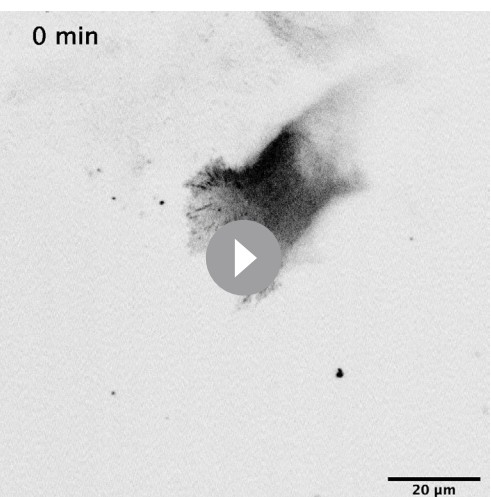

**Video 2.** Focal adhesions dynamics on 35 nm integrin α5β1 ligand lateral spacing. Representative time-lapse fluorescent imaging of a keratinocyte transfected with mCherry-α-paxillin to visualize focal adhesions dynamics during sheet migration.

https://elifesciences.org/articles/69861/figures#video2

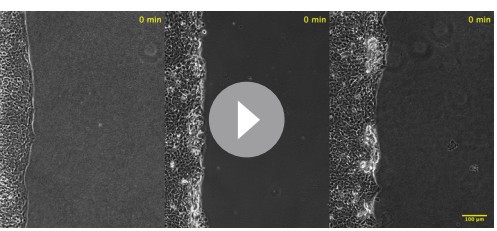

**Video 1.** Keratinocyte sheet migration on 35, 50, and 70 nm integrin α5β1 ligand lateral spacing. Time-lapse phase contrast imaging showing the migratory behaviour over 16 hr.

https://elifesciences.org/articles/69861/figures#video1

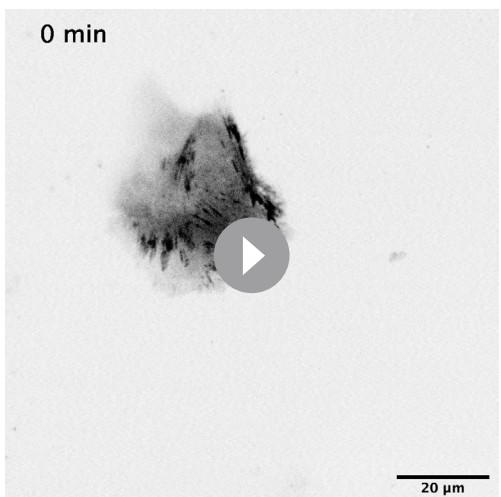

**Video 3.** Focal adhesions dynamics on 50 nm integrin α5β1 ligand lateral spacing. Representative time-lapse fluorescent imaging of a keratinocyte transfected with mCherry-α-paxillin to visualize focal adhesions dynamics during sheet migration.
https://elifesciences.org/articles/69861/figures#video3

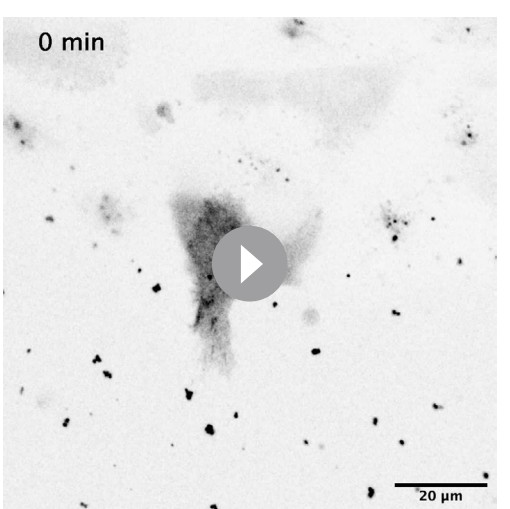

**Video 4.** Focal adhesions dynamics on 70 nm integrin α5β1 ligand lateral spacing. Representative time-lapse fluorescent imaging of a keratinocyte transfected with mCherry-α-paxillin to visualize focal adhesions dynamics during sheet migration.
https://elifesciences.org/articles/69861/figures#video4

focal adhesions exhibited longer lifetimes and therefore slower remodelling (*Figure 2A and B*). This is supported by the quantified average focal adhesions area, which is the smallest in cells on 50 nm compared to 35 or 70 nm inter-ligand spacing (*Figure 2C*), as well as focal adhesion length, where we observe the longest adhesions on 70 nm interparticle spacing (*Figure 2—figure supplement 1C*). Moreover, the diffuse mCherry-α-paxillin signal present in the cytoplasm at the same focal plane of focal adhesions allowed us to quantify the surface contact area of each cell, thus focal adhesion density (*Figure 2—figure supplement 1A*, B). Cells showed the largest surface contact area and density of focal adhesion at 35 nm which both decrease with increasing interparticle spacing in agreement with the previous reports in similar conditions of single cells (*Cavalcanti-Adam et al., 2007*).

Integrin α5β1 is not the sole ligand involved in keratinocyte migration. During wound healing, basal keratinocytes also interact with their provisional basement membrane using αv-containing integrin isoforms. We therefore examined whether migration was regulated in the same manner when engaging other integrins by using the c(RGDfK) peptide for cell attachment (*Figure 1—figure supplement 4A*). c(RGDfK) allows for cell adhesion through a broader range of integrins, including αvβ1 and αvβ6, which are expressed by basal keratinocytes in the wound (*Cavani et al., 1993*; *Kapp et al., 2017*; *Koivisto et al., 2014*; *Sixt et al., 2001*). While c(RGDfK) still allows for α5β1 interaction, the affinity for this integrin subtype is lower compared to αv-containing isoforms ($IC_{50}$ = 200 nM) (*Kapp et al., 2017*; *Figure 1—figure supplement 1B*). Interestingly, on c(RGDfK), keratinocytes exhibit faster migration on 35 nm inter-ligand spacing (vs. 50 nm on α5β1-presenting substrates), and correlation length does not change with receptor densities, which we showed to be biphasic for α5β1 (*Figure 1—figure supplement 4B, C*).

All together, these data reveal the existence of a specific optimal integrin α5β1 density that promotes faster focal adhesion dynamics and more efficient keratinocyte coordination, thereby leading to the most effective re-epithelialization process.

## Integrin α5β1 nanospacing controls force propagation in keratinocyte monolayers

Epithelial cells coordinate their movements as intercellular stress polarization increases, leading to collective cell migration through mechano-transductive processes (*Das et al., 2015*; *Tambe et al., 2011*; *Trepat and Fredberg, 2011*). Integrin nano-presentation in single cells has been shown to regulate not only the persistence of migration but also traction force generation (*Cavalcanti-Adam et al., 2007*; *Oria et al., 2017*). Therefore, we sought to understand how integrin α5β1 density controls the amount of surface traction forces, and consequently, the intercellular

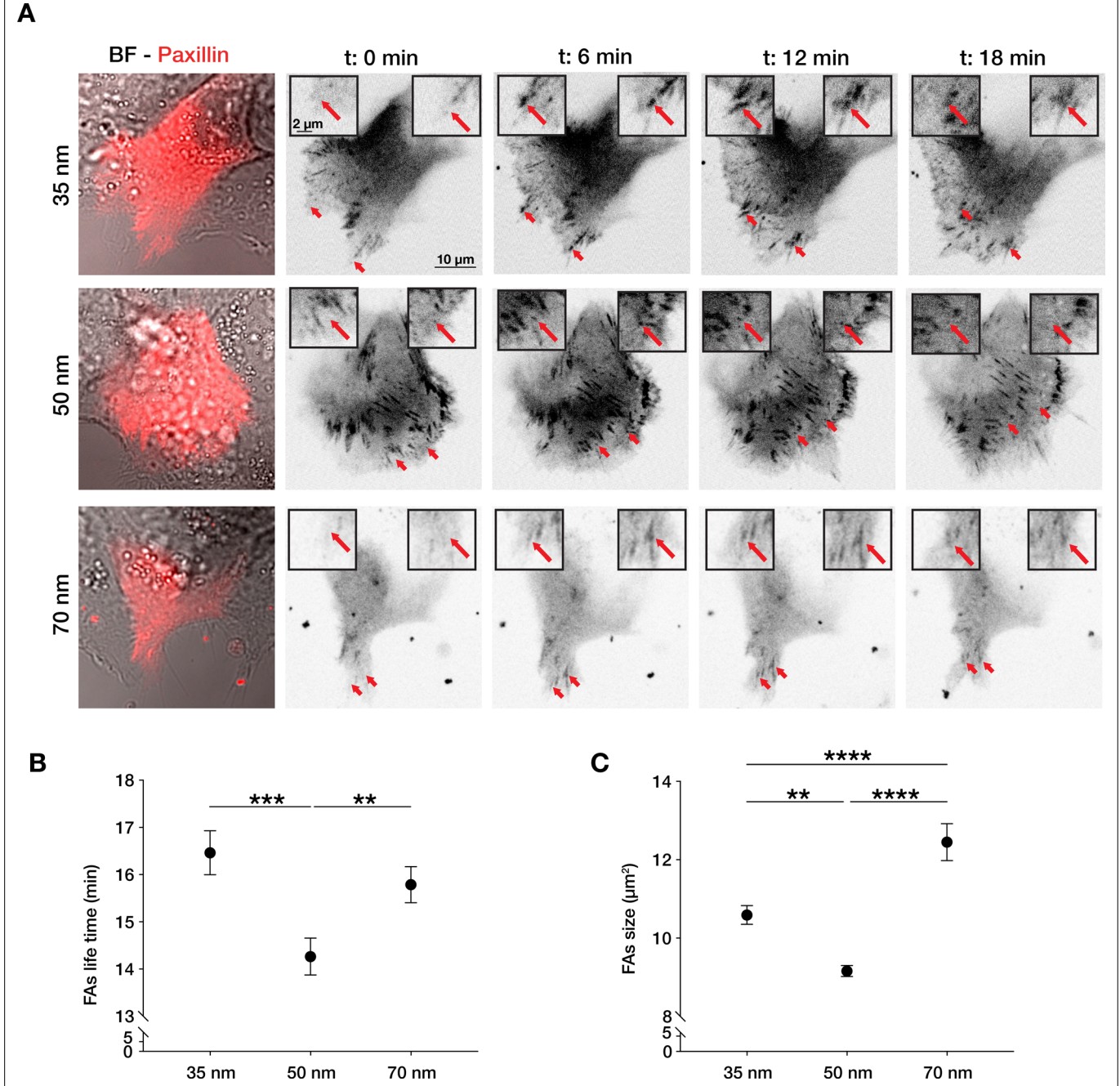

**Figure 2.** Integrin α5β1 lateral spacing governs focal adhesion dynamics and size. (**A**) Representative still frames from time sequence over 18 min from transiently transfected keratinocytes with mCherry-α-paxillin migrating on 35, 50, or 70 nm integrin α5β1 ligand lateral spacing hydrogels. The red arrows and insets highlight the development of the same focal adhesion over culture time. Left column images are of BF = bright-field overlaid with mCherry-α-paxillin (red) at *t* = 0. Other images are of mCherry-α-paxillin in B&W over culture time (*t* = 0, 6, 12, 18 min). (**B**, **C**) Quantification of focal adhesions (FAs) lifetime and size (area, in μm²) on the three different spacing surfaces. The data points show mean values ± s.e.m. of >10 cells/condition, from at least three independent experiments. **p < 0.01; ***p < 0.001; ****p < 0.0001 using a Mann-Whitney test.

The online version of this article includes the following figure supplement(s) for figure 2:

**Source data 1.** Data points for graphs in *Figure 2* and its supplements.

**Figure supplement 1.** Cell adhesion area and focal adhesion density and length depends on ligand spacing.

stresses in the monolayer. To do this, we incorporated fluorescent beads into nanopatterned PAA hydrogels to perform traction force microscopy (TFM) on migrating monolayers. Surprisingly, this revealed a significant increase in monolayer tractions with decreasing integrin α5β1 ligand density (*Figure 3B and C*), which did not follow the optimum migratory behaviour (velocity, correlation length) that we observed on 50 nm inter-ligand spacing (*Figure 1*). We previously showed that the distance of stress propagation within the monolayer is crucial for collective cell migration, and that this is related to the levels of actomyosin contraction in individual cells (*Das et al., 2015*; *Vishwakarma et al., 2018*). Therefore, we were curious if integrin α5β1 lateral spacing differentially regulates the length scale of stress propagation in keratinocyte sheets, independent from the amount of traction forces generated.

To investigate this, we calculated the stress vectors using a force balance algorithm (monolayer stress microscopy [MSM]) (*Tambe et al., 2011*; *Vishwakarma et al., 2018*) from the measured traction forces. We did this within confluent monolayers themselves in order to avoid any bias resulting from migration of the cell sheets (*Figure 3A*). While the comparison of the absolute values of stress was in line with the traction forces, the spatial correlation length of the stress vectors was the largest on 50 nm inter-ligand distance ($C(r)$ ~ 200 μm) vs. on 35 or 70 nm ($C(r)$ ~ 50 μm) (*Figure 3A and B*, *Figure 3—figure supplement 1*). To understand if this stress correlation length relationship was specific to α5β1 integrin, we performed MSM on c(RGDfK)-functionalized substrates. The results show no difference between the force correlation length on the three inter-ligand densities, underlining the specific role of α5β1 in controlling the length of force propagation in keratinocyte monolayers (*Figure 1—figure supplement 4*).

In conclusion, the monolayer force correlation length was in line with the velocity correlation length in migrating keratinocyte sheets, suggesting that keratinocytes have an optimum integrin α5β1 density that best dictates the direction of intercellular stresses, and thereby the extent of force propagation in the monolayer.

## Integrin α5β1 lateral spacing overrides substrate stiffness and controls E-cadherin-mediated collective cell migration efficiency

It has been shown both in vivo and in vitro that collective cell migration is affected by substrate rigidity, with stiffer substrates generally enhancing migration efficiency (*Balcioglu et al., 2020*; *Barriga et al., 2018*; *Ng et al., 2012*). Furthermore, it has been observed that focal adhesions lifetime and the stability of its components increase with increasing substrate stiffness and traction forces (*Fusco et al., 2017*; *del et al., 2009*; *Trichet et al., 2012*; *Zhou et al., 2017*). Therefore, we sought to understand the influence of substrate stiffness on the integrin α5β1 lateral spacing effect previously described on 23 kPa hydrogels. Migration and MSM experiments were performed within an in vivo-relevant range of tissue stiffness, that is, on PAA hydrogels of 11, 23, 55, and 90 kPa (*Goffin et al., 2006*) at the three inter-ligand spacings (35, 50, and 70 nm). Unlike on protein-coated surfaces (*Vishwakarma et al., 2018*), keratinocytes were not able to form cohesive monolayers on the softer (<11 kPa) or stiffer (>90 kPa) nanopatterned surfaces, regardless of spacing. When comparing stiffness alone, higher substrate rigidity induced higher migration speed, as previously reported by others (*Balcioglu et al., 2020*; *Ng et al., 2012*; *Figure 4A*). Interestingly, we observed for each stiffness that cell sheet migratory speed was always more efficient when integrin α5β1 ligand lateral spacing was 50 nm vs. 35 or 70 nm (11 kPa: ~5 μm/hr at 50 nm vs. < ~3 μm/hr at 35 and 70 nm; 23 kPa: ~17 μm/hr vs. < ~9 μm/hr; 55 kPa: ~15 μm/hr vs. < ~10 μm/hr; 90 kPa: ~14 μm/hr vs. < ~9 μm/hr) (*Figure 4A*). The correlation of velocity vectors also confirmed a greater ability of keratinocytes to coordinate migration on the 50 nm inter-ligand spaced substrates (11 kPa: ~300 μm vs. ~200 μm; 23 kPa: ~ 400 μm vs. < ~ 200 μm; 55 kPa: ~400 μm vs. ~200 μm; 90 kPa: ~ 350 μm vs. < ~190 μm) (*Figure 4B*). The extent of force propagation within the monolayers also confirms the higher force-propagation capability when integrin α5β1 ligand has a lateral spacing of 50 nm (*Figure 4C*). MSM could not be carried out reliably on the stiff 90 kPa hydrogels due to limited bead movement caused by the specific surface functionalization, resulting in a high signal-to-noise ratio of the bead displacements. Nevertheless, speed and velocity correlation length observations on 90 kPa hydrogels show the 50 nm trend that we observe on the softer substrates.

Finally, we sought to understand how integrin α5β1 ligand spacing directly regulates intercellular force transmission efficiency in keratinocytes. In epithelial sheets, the actin cytoskeleton physically

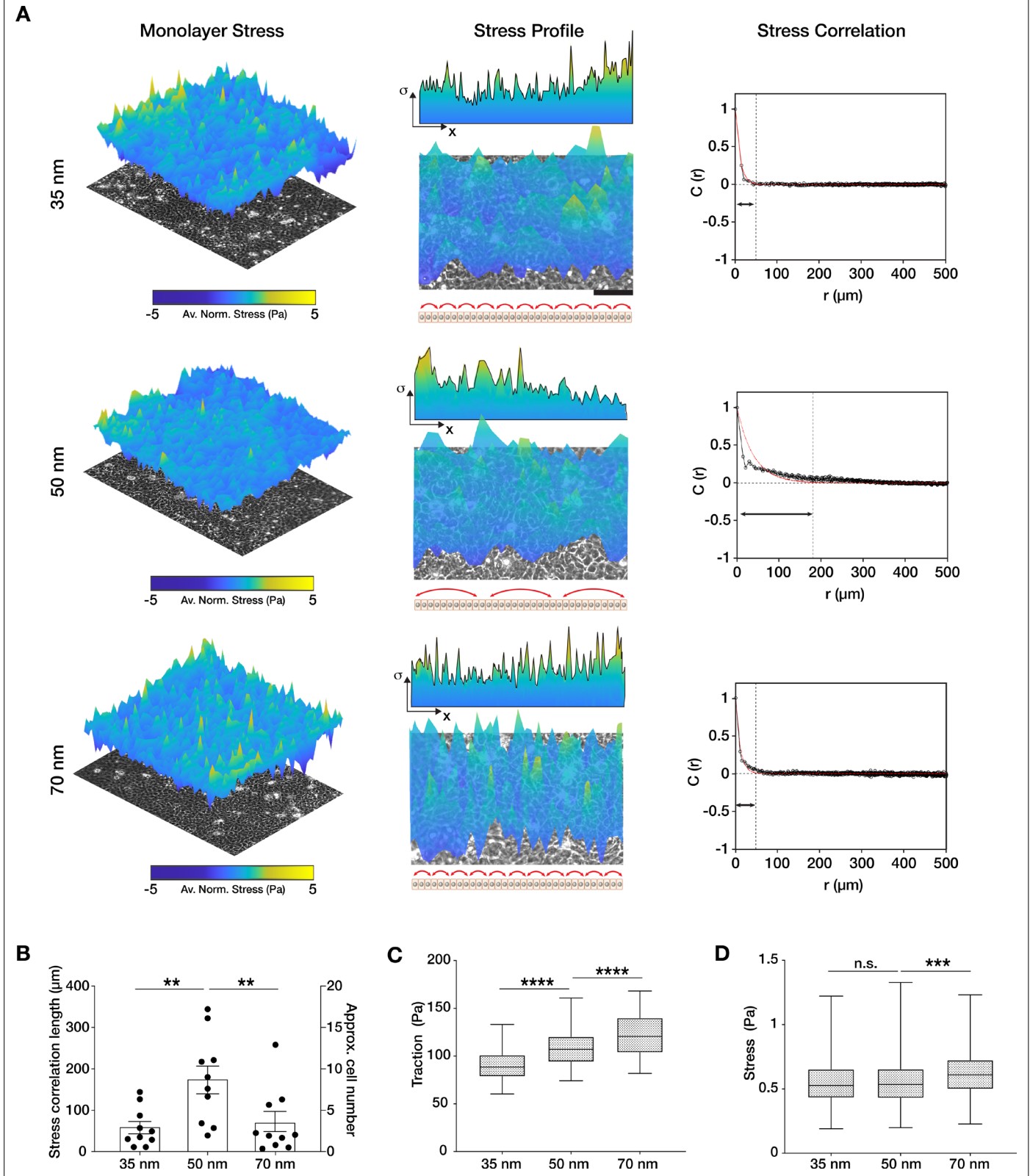

**Figure 3.** Traction force generation depends on integrin α5β1 ligand nanospacing. (**A**) Monolayer stress plots and profiles, and spatial correlation curves from keratinocyte monolayers at the different integrin α5β1 ligand lateral spacings (35, 50, 70 nm). Stress correlation can be identified by inter-peak distance of the stress profile as schematically represented by the red arrows. Scale bar is 100 μm; σ = stress. The quantification of stress correlation lengths (**B**) shows an optimal intercellular force coordination at 50 nm inter-ligand spacing. (**C**) Traction forces and stresses (**D**) quantified in

*Figure 3 continued on next page*

*Figure 3 continued*

the three spacing conditions. Scatter and box and whiskers plots show values and mean ± s.e.m. from at least four independent experiments. n.s. = not significant; **p < 0.01; ***p < 0.001; ****p < 0.0001 using a Mann-Whitney test.

The online version of this article includes the following figure supplement(s) for figure 3:

**Source data 1.** Data points for graphs in *Figure 3* and its supplements.

**Figure supplement 1.** Quantification of stress correlation lengths in human epidermis-derived keratinocytes (hKC) (**A**) and primary human epidermal keratinocytes (nHEK) (**B**) monolayers on integrin α5β1 ligand lateral spacings (35, 50, 70 nm).

connects cell-ECM adhesion structures with cell-cell adhesions (*Figure 1A*). Between the latter, adherens junctions are responsible for maintaining the intercellular force continuum via E-cadherin-based homophilic interactions (*Bazellières et al., 2015*; *Li et al., 2012*; *Ng et al., 2012*). This mechanical continuum is crucial for coordinating collective migration by regulating intercellular pulling forces necessary for mechanotransduction pathways upstream of cell polarization (*Das et al., 2015*). We

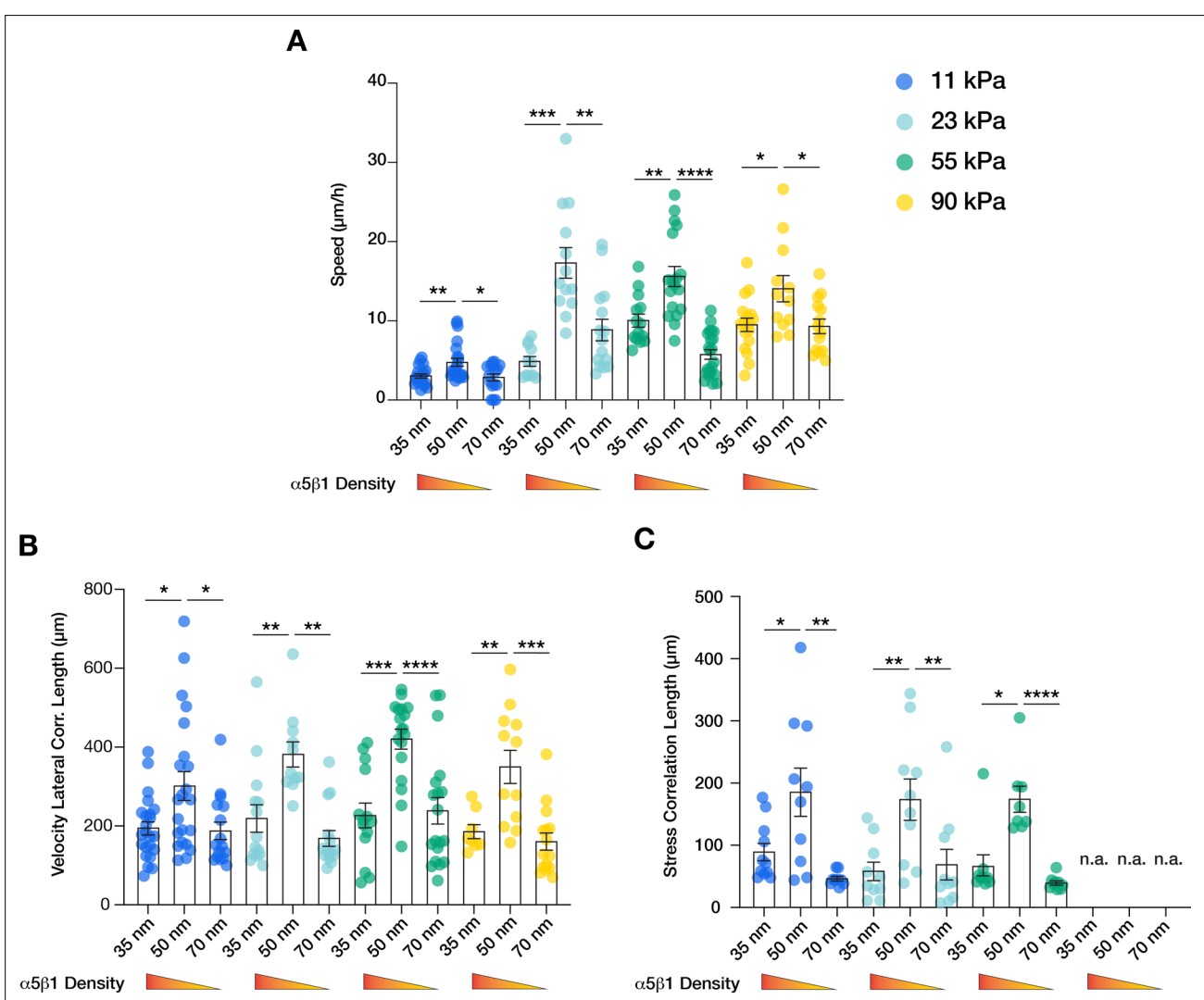

**Figure 4.** α5β1 ligand spacing outweighs substrate stiffness in collective cell migration. Quantification of the migratory speed (**A**), velocity lateral correlation length (**B**), and stress correlation length (**C**) of keratinocyte sheets on different substrate rigidities (11 kPa – blue, 23 kPa – teal, 55 kPa – green, 90 kPa – yellow) and integrin α5β1 ligand lateral spacing (35, 50, 70 nm). Scatter plots show values and mean ± s.e.m. from at least three independent experiments. n.a. = not applicable; *p < 0.05; **p < 0.01; ***p < 0.001; ****p < 0.0001 using a Mann-Whitney test.

The online version of this article includes the following figure supplement(s) for figure 4:

**Source data 1.** Data points for graphs in *Figure 4*.

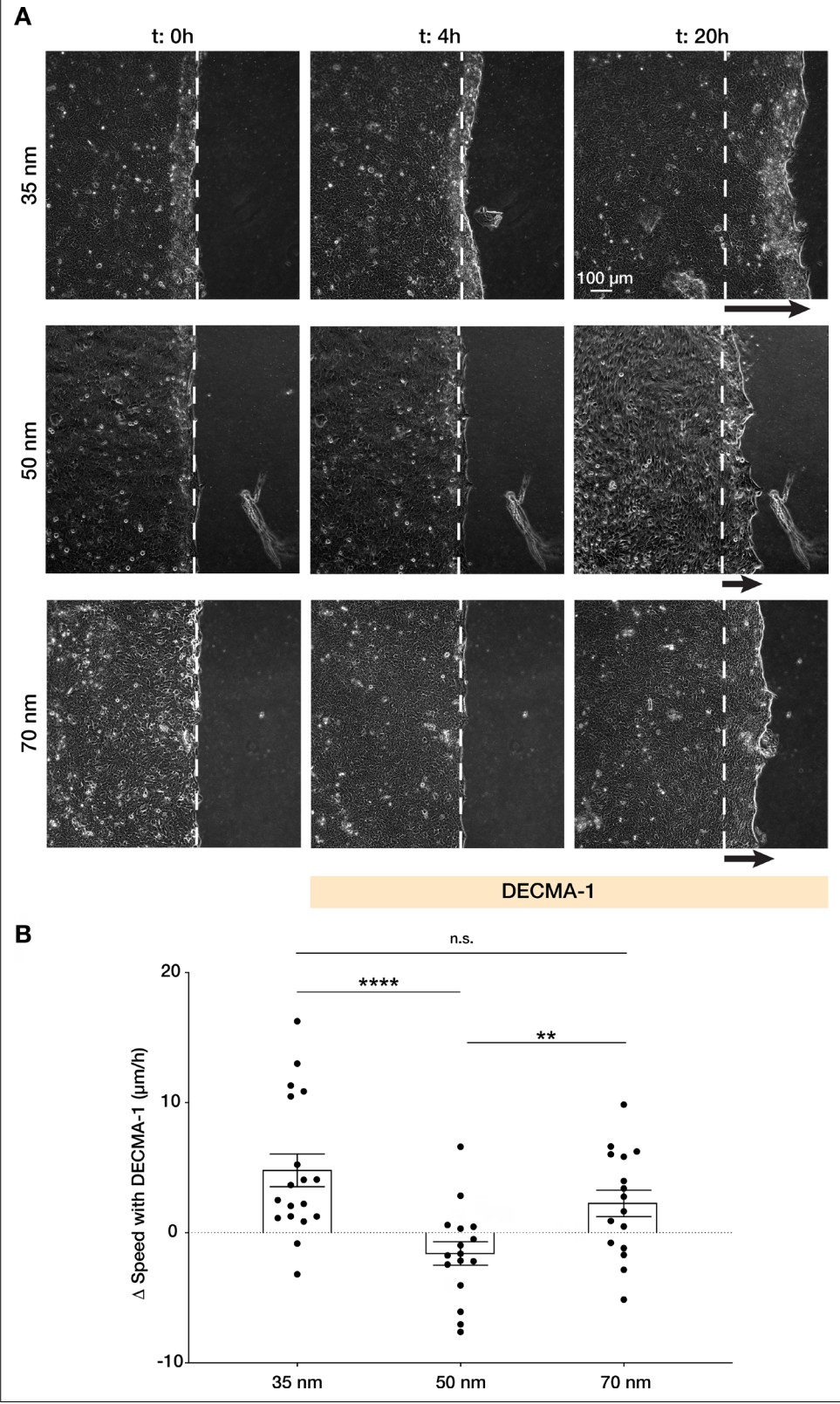

**Figure 5.** Integrin α5β1 spacing regulates intercellular force transmission via E-cadherin. (**A**) Representative frames of keratinocyte sheet migration before (*t* = 0–4 hr) and after (*t* = 4–20 hr) the addition of E-cadherin blocking antibody (DECMA-1) in the medium. The arrows illustrate an approximation of the extent of sheet elongation. (**B**) Quantification of the change in migratory speed in the presence of DECMA-1 vs. control. Scatter plots show values

*Figure 5 continued on next page*

*Figure 5 continued*

and mean ± s.e.m. from at least three independent experiments. n.s. = not significant; **p < 0.01; ****p < 0.0001 using a Mann-Whitney test.

The online version of this article includes the following figure supplement(s) for figure 5:

**Source data 1.** Data points for graph in *Figure 5*.

hypothesized that outside of the optimal 50 nm inter-ligand spacing of α5β1 peptide we observed, single keratinocytes would move counterproductively, thereby inhibiting monolayer motion. Thus, we inhibited E-cadherin interactions with the addition of a specific blocking antibody (DECMA-1) as previously shown (*Das et al., 2015*). When E-cadherin was inhibited, keratinocyte migration was enhanced on both 35 and 70 nm inter-ligand spacing vs. 50 nm, as indicated by the distance of migration over culture time (*Figure 5A*).

Furthermore, when quantifying the change in sheet speed, Δv, after inhibition of E-cadherin vs. untreated cells, we observed migration speed was reduced on 50 nm vs. enhanced on 35 and 70 nm (*Figure 5B*, *Video 5*), indicating that the previously observed migration efficiency on 50 nm requires precise coordination between cell-cell and cell-ECM structures.

All together, these data indicate a vital role of integrin α5β1 lateral spacing in regulating keratinocyte collective migration. The optimal 50 nm lateral spacing outweighs the effect of substrate rigidity, as well as regulates E-cadherin-mediated transmission of intercellular forces.

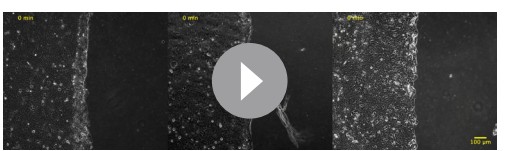

**Video 5.** Keratinocyte sheet migration on 35, 50, and 70 nm integrin α5β1 ligand lateral spacing is enhanced upon E-cadherin blocking. Time-lapse phase contrast imaging showing the migratory behaviour before (1–230 min) and after (240–1240 min) E-cadherin blocking (DECMA-1).

https://elifesciences.org/articles/69861/figures#video5

## Discussion

We show here that ligand nanospacing is integral to collective migration in keratinocytes, even outweighing the effects of substrate stiffness. Specifically, we found that 50 nm inter-ligand distance is optimal for integrin α5β1, whereby collective cell migratory speed (*Figure 1C and E*)

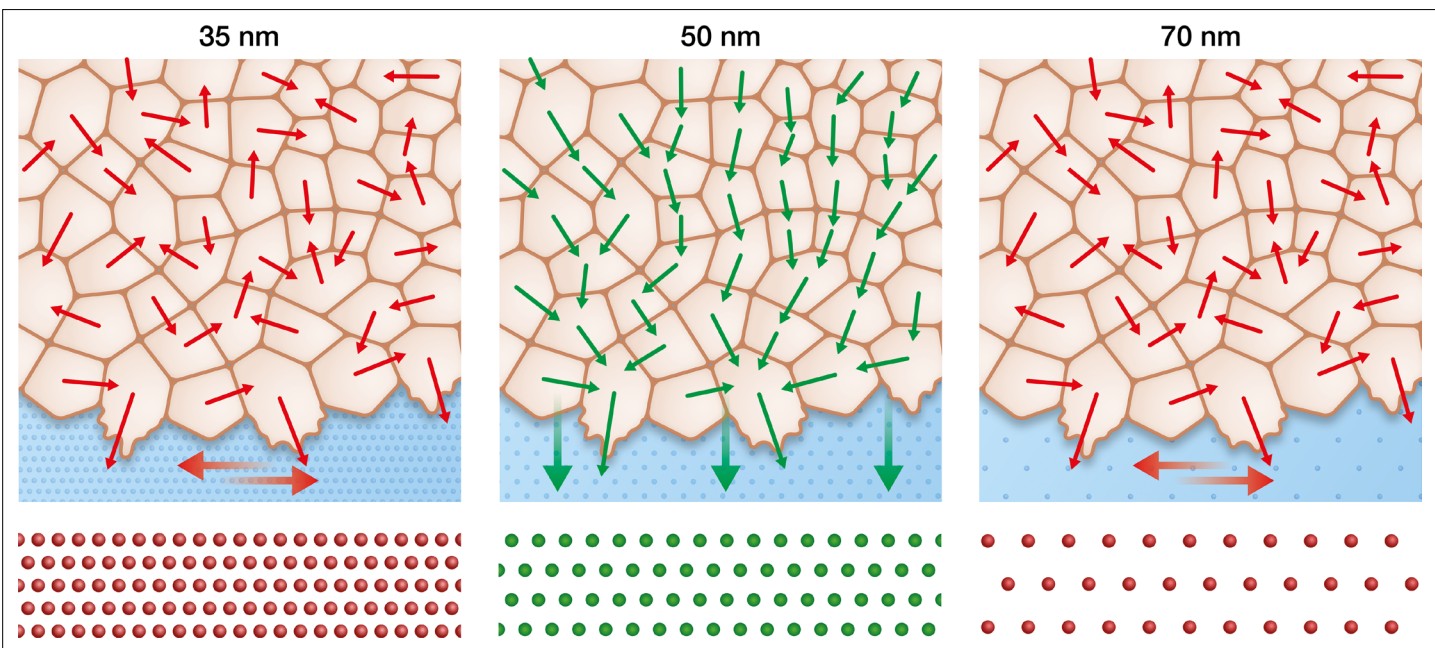

**Figure 6.** Keratinocytes require an optimum integrin α5β1 density to efficiently collectively migrate. In contrast to the optimal integrin α5β1 lateral spacing (green), lower and higher spacings (red) lead to uncorrelated single-cell movement and stress propagation. This results in inefficient collective behaviour, significantly slowing keratinocyte sheet migration.

and coordination were enhanced. At this distance, integrin α5β1 promotes the largest velocity correlation within the monolayer vs. at lower or higher ligand spacing (~400 µm on 50 nm vs. ~200 µm on 35 or 70 nm), indicating that cell movements are transmitted up to 20 cell lengths within the sheet. Consistent with enhanced migration, keratinocytes on this ligand spacing also exhibited the most dynamic focal adhesions, with short lifetimes of ~14 min vs. ~16 min on 35 or 70 nm (*Figures 1D, F and 2*). Quantification of the monolayer stress distribution reveleated that this ideal inter-ligand spacing is not dependent on the absolute traction forces generated on the surface, but rather by the correlation length of stress vectors in the monolayer (*Figure 3A and B*, *Figure 3—figure supplement 1*). This was also confirmed on hydrogels of varying Young's moduli (11–90 kPa), with keratinocytes exhibiting enhanced migratory behaviour (faster and larger velocity and stress correlation lengths) at 50 nm spacing regardless of substrate stiffness (*Figure 4*). This was particularly interesting because higher substrate rigidity is generally known to induce greater cellular traction forces, resulting in faster epithelial sheet migration (*Balcioglu et al., 2020*; *Ng et al., 2012*; *Sunyer et al., 2016*).

Each epithelial cell within a monolayer migrates in the direction of the maximal normal stress in a phenomenon called plithotaxis (*Tambe et al., 2011*; *Trepat and Fredberg, 2011*). In cell collectives, adherens junctions, which are connected to the actin cytoskeleton, are required for intercellular force transmission (*Bazellières et al., 2015*; *Ng et al., 2012*). As the actin cytoskeleton is also connected to the ECM, this explains why we observed disrupted keratinocyte migration on 50 nm integrin α5β1 ligand lateral spacing when E-cadherin interactions were inhibited by addition of DECMA-1 antibody (*Figure 5*). In contrast, at higher and lower ligand spacing, keratinocytes were able to migrate faster when we disturbed intercellular force transmission (*Figure 5*). Therefore, we can conclude that outside of the optimal integrin α5β1 inter-receptor spacing of 50 nm, uncorrelated intercellular stresses slow keratinocyte sheet migration and result in inefficient collective movement (*Figure 6*).

It has previously been shown in single cells that integrin α5β1 regulates the generation of traction forces upon adhesion, whereas αv-class integrins mediate ECM rigidity sensing (*Schiller et al., 2013*). While we do observe faster collective migration with increasing substrate stiffness, as has been previously reported (*Figure 4A*; *Balcioglu et al., 2020*; *Ng et al., 2012*; *Sunyer et al., 2016*), α5β1 inter-receptor spacing outweighs stiffness effects alone. This could be attributed to integrin α5β1's role in regulating traction force generation but not rigidity sensing. Indeed, higher substrate rigidity causes higher traction forces at the sheet edge at the outset of migration (removal of PDMS stencil), thereby transmitting higher net stress to the cell followers, ultimately resulting in more efficient collective migration (*Sunyer et al., 2016*). This validates our data that 50 nm lateral spacing of integrin α5β1 is required for efficient force transmission independent from the net force exerted on the cells.

The severe phenotype acquired by murine animal models genetically lacking α5 or β1 integrin subunits has prevented our deep understanding into the roles specific integrin subtypes and organization play in wound healing (*Raghavan et al., 2000*; *Yang et al., 1993*). With our bottom-up approach, we systematically identified the importance of integrin α5β1 density in coordinating force propagation within keratinocyte monolayers. How this inter-ligand optimal spacing can be understood in the in vivo context still requires further exploration, but our findings suggest that an integrin subtype-specific arrangement may be present in specific biological contexts that function through mechanosensitive pathways mediated by focal adhesions. Future studies will be required to address this phenomenon in order to more closely dissect the role of integrins and associated signaling pathways in the mechanobiology of wound closure.

## Materials and methods

### Key resources table

| Reagent type (species) or resource | Designation | Source or reference | Identifiers | Additional information |
|---|---|---|---|---|
| Cell line (human) | HaCaT | Cell Lines Service – CLS, Germany | 300,493 | |
| Cell line (*Homo-sapiens*) | hKC | *Sawant et al., 2018* | – | Gift from Dr Rudolf Leube |
| Cell line (*Homo-sapiens*) | nHEK | CellSystems | FC-0064 | |

*Continued on next page*

*Continued*

| Reagent type (species) or resource | Designation | Source or reference | Identifiers | Additional information |
|---|---|---|---|---|
| Recombinant DNA reagent | mCherry-$\alpha$-Paxillin-C1 (plasmid) | *Efimov et al., 2008* | | Gift from Dr Irina Kaverina |
| Antibody | Anti-human E-cadherin (DECMA-1, rat monoclonal) | Millipore | MABT26 | (10 µg/ml) |
| Peptide, recombinant protein | Thiolated integrin $\alpha$5β1 peptidomimetic | *Fraioli et al., 2015*; *Guasch et al., 2015*; *Neubauer et al., 2013* | – | – |
| Peptide, recombinant protein | Thiolated c(RGDfK) | PSL GmbH, Heidelberg, Germany | customized | – |
| Commercial assay or kit | Click-iT Plus EdU imaging kit | Invitrogen | C10640 | – |
| Software, algorithm | MATLAB | MathWorks | – | – |
| Software, algorithm | Imaris | Bitplane, Oxford Instrument | – | – |
| Software, algorithm | ImageJ | NIH | – | – |
| Other | DAPI stain | Invitrogen | D1306 | 1 µg/ml |

## Au-nanopatterned glass surfaces preparation

To obtain PAA nanopatterned surfaces, the desired Au-nanoparticle pattern was first obtained on 18 mm diameter glass coverslips (Carl Roth, Germany) as previously described (*Arnold et al., 2004*; *Cavalcanti-Adam et al., 2007*). Briefly, the coverslips were cleaned in piranha solution ($3H_2SO_4$:$1H_2O_2$), rinsed thoroughly with MilliQ water and stored in a dust-free environment until further use. Au-micellar solution was obtained first by dissolving a diblock copolymer polystyrene-b-poly(2-vinylpyridine) (Polymer Source) in *o*-xylene and second by loading with $HAuCl_4 \cdot 3H_2O$ (Sigma Aldrich) according to the specific parameter $L = n[HAuCl_4]/n[P2VP]$. The prepared micellar solution was used to spin-coat the clean coverslips which were subsequently treated with argon-hydrogen plasma (90 % Ar/10% $H_2$) in a Tepla PS210 microwave plasma system (PVA Tepla, Germany) for 45 min at 200 W and 0.4 mbar in order to remove the polymer. Different ratios of polystyrene/poly(2-vinylpyridine) units (288/119 or 501/323) and spin speeds (3000–8000 rpm) were employed to obtain 35, 50, and 70 nm interparticle spacing. Nanopatterned glass surfaces were then evaluated by scanning electron microscopy (SEM) (Carl Zeiss, Germany) as previously described (*Arnold et al., 2004*; *Cavalcanti-Adam et al., 2007*). Interparticle spacing and overall order was quantified with the *k*-nearest-neighbours algorithm implemented by an in-house script written in ImageJ software. The *k*-nearest neighbours (*k* = 6 for ordered particles) was estimated for more than 600 particles per sample.

## Transfer of Au-nanopatterns onto PAA hydrogels

To be able to transfer the gold nanoparticles from the glass surface to the hydrogels while preserving their distribution, the nanopatterned coverslips were incubated with 0.5 mg/ml *N,N'*-bis(acryloyl) cystamine (Thermo Fisher) in ethanol for 1 hr at room temperature, washed with ethanol, and dried with nitrogen stream. Afterward, hydrogels were formed by pipetting a mixture of acrylamide (Bio-Rad), bis-acrylamide (Bio-Rad), 0.003 % tetramethylethylenediamine (Bio-Rad), and 0.03 % ammonium persulfate (Sigma) diluted in phosphate-buffered saline (PBS) between glutaraldehyde-activated glass bottom dishes (MatTek) and AuNP-functionalized coverslips. The following ratio of acrylamide/bis-acrylamide were employed to reach different hydrogel Young's moduli: 10%/0.07 % for 11 kPa; 7.5%/0.2 % for 23 kPa; 12%/0.6 % for 55 kPa; 12%/0.3 % for 90 kPa (*Aratyn-Schaus et al., 2010*). Hydrogels were allowed to swell for at least 72 hr in PBS to facilitate the detachment of the coverslips and the transfer efficiency was evaluated by SEM of both the glass and hydrogels surface. The obtained hydrogels were sterilized with 30 min ultraviolet light irradiation before their employment as cell culture substrates.

## Cell culture

HaCaT (Cell Lines Service – CLS, Germany) were cultured with Dulbecco's modified Eagle medium (Gibco, Thermo Fisher Scientific) supplemented with 10 % fetal bovine serum (Sigma Aldrich) and 1 % penicillin-streptomycin (Gibco, Thermo Fisher Scientific) at 5 % $CO_2$ and 37°C . To passage and perform experiments, cells were detached with 10 min incubation with 5 mM EDTA solution in PBS followed by the incubation with 0.05 % trypsin/EDTA (Gibco, Thermo Fisher Scientific). hKC cells were kindly provided by Dr Rudolf Leube. The cells were grown at 37 °C and 5 % $CO_2$ in EpiLife medium with human keratinocyte growth supplement (Gibco, Grand Island, NY) and penicillin/streptomycin. Cells were detached using accutase (Sigma) for 5 min at 37 °C after incubation with 5 mM EDTA solution in PBS and resuspended in trypsin neutralizing solution (Gibco, Thermo Fisher Scientific) before use. Primary human keratinocytes (nHEK, CellSystems) were cultivated according to the manufacturer's instructions for a maximum of three passages. Cell lines and primary cells were routinely tested negative for mycoplasma infection.

## Migration experiments

To perform cell migration experiments, nanopatterned hydrogels were incubated for 2 hr with 25 μM customized thiolated c(RGDfK) peptide (PSL GmbH, Heidelberg, Germany) (*Haubner et al., 1996*; *Figure 1—figure supplement 1B*) or with thiolated integrin α5β1 peptidomimetic (*Fraioli et al., 2015*; *Guasch et al., 2015*; *Neubauer et al., 2013*; *Figure 1—figure supplement 1A*) to allow for cell adhesion. Bovine serum albumin-coated PDMS stencils were employed to horizontally confine the cells on the surface and obtain a confluent monolayer of approximately 3 × 6.5 mm. Cells were seeded with a density of 4500 cells per mm², allowed to adhere and to reach confluency for approximately 12  hr at 5 % $CO_2$ and 37° C, and triggered to migrate with the lift-off of the confinement. In the case of hKC and nHEK experiments, seeding and experimental media were supplemented with 1.2 mM calcium chloride (Sigma). Cell migration experiments were carried out either inside a stand-alone incubator or within an incubator staged over an Axio Observer 7 (Carl Zeiss, Germany). Images were acquired every 10 minutes at multiple positions using an automated stage controlled by Zen software (Carl Zeiss, Germany). For E-cadherin blocking experiments, cell monolayers were allowed to migrate for at least 4 hr before exchanging the medium with pre-warmed medium containing 10 μg/ml E-cadherin blocking antibody (clone DECMA-1, Millipore MABT26).

## TFM and MSM

TFM and MSM were performed as previously described (*Das et al., 2015*; *Vishwakarma et al., 2018*); 1 μm fluorescent carboxylate-modified polystyrene beads (Sigma) were mixed in the PAA solution before polymerization and allowed to reach the hydrogel surface by gravity during gelation. Cell-induced bead displacement vectors were calculated by comparing the picture with relaxed beads (cell-free gel after trypsinization) from the picture acquired with cells using the PIV plugin of ImageJ. Traction forces were calculated from these vectors using the Fourier transform traction cytometry plugin. Average normal stress vectors within the monolayer were calculated using the traction force information using a force balance algorithm in MATLAB (MathWorks) as formulated elsewhere (*Vishwakarma et al., 2018*). The force propagation within the monolayer is characterized by the force correlation length, which is the length scale of the following spatial autocorrelation function:

$$C\left(r\right) = \frac{1}{N var(\sigma')} \sum_{i,j=1}^{N} \sum_{|r_i - r_j|=r} \delta\sigma_i' \cdot \delta\sigma_j'$$

where $\delta\sigma'$ are the local deviation of the average normal stress at position from its spatial mean $\sigma'$ and var $(\sigma')$ is its variance. The correlation length of the stresses was determined at the point where the function was equal to 0.01 (*Vishwakarma et al., 2018*).

## Focal adhesion and cell surface contact area quantification

To be able to visualize focal adhesion dynamics and quantify their lifetime, HaCaTs were transfected with mCherry-α-Paxillin-C1 construct kindly provided by Dr Irina Kaverina (*Efimov et al., 2008*). Transient transfection was performed following the Fast-Forward protocol provided by the Attractene transfection reagent kit (Qiagen, Germany). Briefly, 0.1 μg of the plasmid per sample was allowed to react with the Attractene reagent for 15 min at room temperature and then mixed with the cell

suspension for cell seeding within the PDMS stencil placed on the hydrogel. After 12 hr, cell migration was initiated and mCherry signal was acquired every 2 min using an LSM 880 confocal microscope (Carl Zeiss, Germany) equipped with a 40× long-distance water immersion objective with a numerical aperture of 1.1. The obtained time-lapse videos were analysed using Imaris image analyses software (Bitplane, Oxford Instrument) automatically tracking each focal adhesion in the cells and quantified its lifetime area and length. The surface contact area of each cell was quantified by segmenting the mCherry signal in the cytoplasm at the focal plane where focal adhesions were visible. Focal adhesions density per cell was quantified by dividing the number of focal adhesions per the corresponding surface contact area.

## Quantification of velocity vectors and correlation length

Velocity vectors and their correlation length, which quantify the ability of cells to coordinate their movements, were determined as previously described (*Das et al., 2015*). Briefly, time-lapse videos obtained using phase-contrast microscopy and two consecutive images with an interval of 10 min were used to calculate velocity vectors using the PIV plugin of ImageJ software. In PIV analyses, each image was broken down in 32 × 32 pixels windows for comparison and a two component ($i,j$) velocity vector was assigned to the centre of the window, namely lateral ($U_{ij}$, perpendicular to the direction of monolayer migration) and axial ($V_{ij}$, along the monolayer migration). The fluctuations of the lateral component of the vectors ($v_{ij}$) were determined as follows:

$$v_{ij} = V_{ij} - \sum_{i=1,m} \sum_{j=1,n} \frac{V_{ij}}{m \times n} = V_{ij} - V_{mean}$$

where $V_{mean}$ is the mean lateral velocity. The lateral velocity correlation function was formulated as follows:

$$C(r) = \langle v(\mathbf{r}') \times v(\mathbf{r}' + \mathbf{r}) \rangle_{\mathbf{r}'} / \left[ \left\langle v(\mathbf{r}')^2 \right\rangle \times \left\langle v(\mathbf{r}' + \mathbf{r})^2 \right\rangle \right]^{1/2}$$

where $\langle \ldots \rangle$ symbolizes the average, $\mathbf{r}$ is the vector of coordinate ($i,j$), and $r$ is the norm of $\mathbf{r}$ defined as r = ‖**r**‖. Similar to the force correlation length, the lateral correlation length was determined at the point where the function was equal to 0.01 (*Das et al., 2015*).

## Quantification of cell density and proliferation

To quantify keratinocyte cell density on the different substrates, the monolayers were fixed for 10 min using 4 % paraformaldehyde in PBS for the same conditions as in migration experiments. Upon permeabilization using 0.3 % Triton X-100 for 2 min, the monolayers were stained for DNA using DAPI. The number of nuclei was counted using ImageJ software (NIH) in immunofluorescent imaged obtained at 10× magnification. Cell proliferation was quantified after 6 hr incorporation of 5-ethynyl-2′-deoxyuridine (EdU) in lateral-confined HaCaT monolayers. EdU staining was performed according to the manufacturer's protocol (Click-iT Plus EdU imaging kit, Invitrogen) together with DAPI to quantify the ratio of proliferating cells. The number of proliferating nuclei was determined using ImageJ software (NIH).

### Statistics

Statistical tests and graphics were performed using GraphPad Prism software, choosing parametric or nonparametric tests after evaluating the normality distribution of the data. Sample conditions and the specific test used for each data set are indicated in corresponding figure captions.

## Acknowledgements

We thank the general support of the Max Planck Society and the grant to JDR from the Interdisciplinary Centre for Clinical Research within the faculty of Medicine at the RWTH Aachen University. JPS acknowledges funding from the Deutsche Forschungsgemeinschaft (DFG, German Research Foundation) under Germany's Excellence Strategy via the Excellence Cluster 3D Matter Made to Order (EXC-2082/1–390761711) and the Gottfried Wilhelm Leibniz Award. JLY acknowledges the support by the Ministry of Education under the Research Centres of Excellence program through the Mechanobiology Institute at the National University of Singapore and the Biomedical Engineering Department

at the National University of Singapore. We thank Adam Breitscheidel for his support with the graphic design.

## Additional information

### Funding

| Funder | Grant reference number | Author |
|---|---|---|
| RWTH Aachen University | Interdisciplinary Centre for Clinical Research | Jacopo Di Russo Timmy Steins |
| Deutsche Forschungsgemeinschaft | EXC-2082/1 - 390761711 | Joachim P Spatz |
| Mechanobiology Institute, Singapore | | Jennifer L Young |
| Max Planck Institute for Dynamics of Complex Technical Systems Magdeburg | | Jacopo Di Russo Jennifer L Young Julian WR Wegner |

The funders had no role in study design, data collection and interpretation, or the decision to submit the work for publication.

### Author contributions

Jacopo Di Russo, Conceptualization, Data curation, Formal analysis, Funding acquisition, Investigation, Methodology, Project administration, Software, Supervision, Validation, Visualization, Writing - original draft, Writing – review and editing; Jennifer L Young, Investigation, J.L.Y. designed and prepared the nanopatterned hydrogels., Methodology, Resources, Validation, Writing – review and editing; Julian WR Wegner, Investigation, Methodology, Validation, Writing – review and editing; Timmy Steins, Investigation, Writing – review and editing; Horst Kessler, H.K. provided the peptidomimetics., Investigation, Resources, Writing – review and editing; Joachim P Spatz, Conceptualization, Funding acquisition, Supervision, Writing – review and editing

### Author ORCIDs

Jacopo Di Russo (ID) http://orcid.org/0000-0001-6731-9612

### Decision letter and Author response

Decision letter https://doi.org/10.7554/eLife.69861.sa1
Author response https://doi.org/10.7554/eLife.69861.sa2

## Additional files

### Supplementary files

• Transparent reporting form

### Data availability

All data generated or analysed during this study are included as supporting file. The source data file has been provided.

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
