## [Decision Letter]

**Acceptance summary:**

The authors show that optimal alpha5beta1 integrin ligand nanospacing improves collective movement of keratinocytes independent of substrate rigidities. The authors also propose that this optimal ligand spacing depends on intercellular stress propagation/co-ordination mediated by E-cadherin. This papers demonstrates for the first time that nanospacing represents a key parameter of keratinocyte sheet migration speed in 2D and hence, for skin wound healing.

**Decision letter after peer review:**

Thank you for submitting your article "Integrin α5β1 nano-presentation regulates collective keratinocyte migration independent of substrate rigidity" for consideration by *eLife*. Your article has been reviewed by 3 peer reviewers, including Reinhard Fässler as Reviewing Editor and Reviewer #1, and the evaluation has been overseen by Jonathan Cooper as the Senior Editor. The following individual involved in review of your submission has agreed to reveal their identity: Frank Schnorrer (Reviewer #2).

Collective cell motion plays important roles in embryogenesis, wound healing and for cancer progression. Speed and directionality is modulated by numerous parameters including gradients of soluble biochemicals and ECM substrates, substrate rigidity, etc. In the present study, Russo and colleagues ad a new parameter, namely integrin ligand nanospacing.

The authors investigated how collective migration of a keratinocyte cell line is influenced by alpha5beta1 integrin ligand nanospacing at different substrate rigidities, and found that the collective movement of keratinocytes (faster focal adhesion dynamics, better keratinocyte co-ordination) improves at optimal alpha5beta1 integrin nanospacing independent of substrate stiffness. In addition, the authors showed the migration efficiency by optimal ligand spacing depends on intercellular stress propagation/co-ordination mediated by E-cadherin. The strength of the paper is the novelty, i.e. the first demonstration that nanospacing represents a key parameter of cell sheet migration speed. The weakness is the preliminary analysis, e.g. of integrin adhesions (lifetime analyses of integrin adhesions, cell-substrate contact) and of only one cell type (keratinocytes).

Essential revisions:

(1) More comprehensive analysis of integrin-based adhesion dynamics and cell area dynamics.

The differences concerning the life-time of integrin adhesions, although statistically significant, are not so high. In addition, there are many other information in the time-lapses which are not used by the authors and which are probably very relevant to reinforce the message of the manuscript. The authors should quantify the size distribution of integrin-based adhesions, the fraction of the cell surface corresponding to integrin-based adhesions. These parameters may be different depending on the experimental conditions, which will provide valuable knowledge for the present manuscript. Note that in the article from Pere Roca-Cusachs laboratory (Oria et al., Nature 2017), which uses the same experimental strategy on isolated cells, they quantify integrin-based adhesion size (among other parameters). A more comprehensive analysis of integrin-based adhesion dynamics would allow comparing the results of the present manuscript with previously published data.

Integrin ligand spacing as well as substrate rigidity are important parameters in defining cell spreading in isolated cells (e.g., Cavalcanti-Adam et al., Byophys J 2007; Giannone et al., Cell 2004; Pelham and Wang, PNAS 1997). Therefore, it would be critical to quantify the cell surface in contact with the substrate in the different experimental conditions.

(2) Demonstrate that the spacing phenomenon also operates in other cell types. An additional cell type would also allow comparing alphaV-beta3 (which is not expressed on keratinocytes) with alpha5-beta1 integrins.

(3) Exclude that proliferation is unaffected by ligand spacing.

---

## [Author Response]

Essential revisions:(1) More comprehensive analysis of integrin-based adhesion dynamics and cell area dynamics.The differences concerning the life-time of integrin adhesions, although statistically significant, are not so high. In addition, there are many other information in the time-lapses which are not used by the authors and which are probably very relevant to reinforce the message of the manuscript. The authors should quantify the size distribution of integrin-based adhesions, the fraction of the cell surface corresponding to integrin-based adhesions. These parameters may be different depending on the experimental conditions, which will provide valuable knowledge for the present manuscript. Note that in the article from Pere Roca-Cusachs laboratory (Oria et al., Nature 2017), which uses the same experimental strategy on isolated cells, they quantify integrin-based adhesion size (among other parameters). A more comprehensive analysis of integrin-based adhesion dynamics would allow comparing the results of the present manuscript with previously published data.

We thank the reviewer for this comment and agree that a more thorough analysis of integrin and cell dynamics would strengthen the paper. As suggested, we have now included more analyses of focal adhesions in the manuscript, including their average size (Figure 2C), density (Figure 2—figure supplement 1B), and length (Figure 2—figure supplement 1C). The data are presented on page 5.

Integrin ligand spacing as well as substrate rigidity are important parameters in defining cell spreading in isolated cells (e.g., Cavalcanti-Adam et al., Byophys J 2007; Giannone et al., Cell 2004; Pelham and Wang, PNAS 1997). Therefore, it would be critical to quantify the cell surface in contact with the substrate in the different experimental conditions.

We quantified the average cell area in direct contact with the hydrogels by adding the total area of mCherry-paxillin expressed per cell in each condition (Figure 2—figure supplement 1A). The data show that cell contact area decreases with increasing spacing as presented on page 5.

(2) Demonstrate that the spacing phenomenon also operates in other cell types. An additional cell type would also allow comparing alphaV-beta3 (which is not expressed on keratinocytes) with alpha5-beta1 integrins.

This work aimed to specifically understand the influence of nano-presentation on keratinocytes migration in a model of wound healing. We chose to address the epidermal wound healing process due to the 2D migration and the interaction with its ECM in the wound, which can be well modelled in-vitro using nanopatterned hydrogels. The role of integrin ligand nano-presentation can be highly different in 2D vs. 3D migration (Doyle A.D. et al., Nat. Communication 2015; Kubow K. E. et al., Current Biology 2013). Hence, we do not think it is applicable to this work to use other epithelial lines (e.g., MDCK or HUVEC), which physiologically migrate in a 3D environment and may interact with the ECM with different mechanisms. Nevertheless, we agreed with the reviewer’s concern about using exclusively one cell line for this study and have therefore, we repeated our experiments using two more keratinocyte cell models, one being a cell line (immortalized human keratinocyte, hKC) and the other being primary keratinocytes (nHEK). The data are now presented in Figures 1, 1—figure supplement 3 and 3—figure supplement 1 and discussed in page 4.

Using these two additional keratinocyte cell models have reinforced our initial data with HaCaTs, in which we find the same spacing-dependent phenomena for migration and stress correlation. Therefore, we have conclusively demonstrated that this 50 nm spacing phenomenon of the alpha5-beta1 receptor is relevant for keratinocyte 2D sheet migration. The inclusion of this work with other epithelial cell lines, which physiologically migrate in 3D environments, or the comparison of integrin alphaV-beta3 function/other integrin subunits are out of the scope of the manuscript and will be the focus of future work of the group.

(3) Exclude that proliferation is unaffected by ligand spacing.

This is important to consider, so we quantified cell proliferation with 6h EdU incorporation, but found no significant differences between the conditions. Thus, we can exclude spacing-dependent proliferation effects in regulating the migration response. The data are presented in page 2 and in Figure 1—figure supplement 2A, C.